# The Human Soluble NKG2D Ligand Differentially Impacts Tumorigenicity and Progression in Temporal and Model-Dependent Modes

**DOI:** 10.3390/biomedicines12010196

**Published:** 2024-01-16

**Authors:** Anthony V. Serritella, Pablo Saenz-Lopez Larrocha, Payal Dhar, Sizhe Liu, Milan M. Medd, Shengxian Jia, Qi Cao, Jennifer D. Wu

**Affiliations:** 1Department of Hematology/Oncology, Feinberg School of Medicine, Northwestern University, Chicago, IL 60611, USA; anthony.serritella@nm.org; 2Department of Urology, Feinberg School of Medicine, Northwestern University, Chicago, IL 60611, USA; saenzlop@gene.com (P.S.-L.L.); payaldhar2019@u.northwestern.edu (P.D.); sizhe.liu@northwestern.edu (S.L.); milanmedd@gmail.com (M.M.M.); s-jia2@northwestern.edu (S.J.); qi.cao@northwestern.edu (Q.C.); 3Department of Microbiology and Immunology, Feinberg School of Medicine, Northwestern University, Chicago, IL 60611, USA

**Keywords:** soluble NKG2D ligands, tumor cell stemness, tumorigenicity, NK cells, tumor immunity

## Abstract

NKG2D is an activating receptor expressed by all human NK cells and CD8 T cells. Harnessing the NKG2D/NKG2D ligand axis has emerged as a viable avenue for cancer immunotherapy. However, there is a long-standing controversy over whether soluble NKG2D ligands are immunosuppressive or immunostimulatory, originating from conflicting data generated from different scopes of pre-clinical investigations. Using multiple pre-clinical tumor models, we demonstrated that the impact of the most characterized human solid tumor-associated soluble NKG2D ligand, the soluble MHC I chain-related molecule (sMIC), on tumorigenesis depended on the tumor model being studied and whether the tumor cells possessed stemness-like properties. We demonstrated that the potential of tumor formation or establishment depended upon tumor cell stem-like properties irrespective of tumor cells secreting the soluble NKG2D ligand sMIC. Specifically, tumor formation was delayed or failed if sMIC-expressing tumor cells expressed low stem-cell markers; tumor formation was rapid if sMIC-expressing tumor cells expressed high stem-like cell markers. However, once tumors were formed, overexpression of sMIC unequivocally suppressed tumoral NK and CD8 T cell immunity and facilitated tumor growth. Our study distinguished the differential impacts of soluble NKG2D ligands in tumor formation and tumor progression, cleared the outstanding controversy over soluble NKG2D ligands in modulating tumor immunity, and re-enforced the viability of targeting soluble NKG2D ligands for cancer immunotherapy for established tumors.

## 1. Introduction

NKG2D, an activating receptor expressed by all human NK cells, also defined as a co-stimulatory receptor for human NKT, CD8T and γδT cells [1,2,3,4,5,6]. In mice, NKG2D is expressed by all NK cells, but is only expressed by activated CD8 T cells [1,2,3,4,5,6]. The function and major signaling pathways of NKG2D in human and mouse NK cells are largely conserved. However, identified natural NKG2D ligands in human and mice are diverse. In humans, identified NKG2D ligands include the MICA/B and HCMV UL-16 binding proteins ULBP1-6 [7,8,9]. These ligands are often expressed by tumor cells and viral infected cells in response to genotoxic insults or viral infection but are generally absent in normal tissues except in gut epithelium at a threshold level [5,10,11,12,13]. The MICA/B molecules are found in almost all human solid tumors, whereas the expression of ULBPs in human tumors are sporadic [7,8,9]. Homologs of human ULBPs were found to be expressed by mice; however, a homolog of human MICA or MICB was not identified in mice. It was demonstrated that mouse or human tumors with forced expression of membrane-bound NKG2D ligands can be rejected in vivo through activation of NK cells and sometimes CD8 T cells, heightening the significance of NKG2D stimulating in anti-tumor immunity [14,15]. However, the majority of human solid tumors express high levels of NKG2D ligands, in particular MICA and MICB molecules, suggesting that tumors developed mechanisms to evade NKG2DL-NKG2D mediated anti-tumor immunity.

Advanced human tumors release soluble NKG2D ligands (sNKG2D-L) via proteolytic pathways, and in some instances via exosome pathways [7,8,9,16]. It has been shown at large that human sNKG2D-L is immune suppressive. Clinical correlative studies have shown elevated serum levels of sMICA and sMICB correlated with disease progression, metastasis, less responsiveness to therapy, and ultimately poor clinical outcome in a broad spectrum of cancer patients with solid tumors [17,18,19,20,21,22,23,24,25]. In preclinical models where human MIC was overexpressed in mouse tumors, eliminating the effect of sMIC by an antibody-clearing serum sMIC or by an antibody-inhibiting MIC shedding, stimulated NK cell and/or CD8T cell anti-tumor immunity and significantly inhibited tumor progression or growth [26,27,28,29,30,31]. These clinical and pre-clinical studies support the notion that soluble human NKG2D ligands, at least sMICA and sMICB, are immune suppressive and that sMIC (A/B) is a cancer therapeutic target. In contrast to these studies, the soluble mouse NKG2D ligand, sMULT-1, was shown to stimulate antitumor immunity and prevent B16F10 melanoma tumor formation [32]. These inconsistencies in different experimental settings raised the question as to whether sNKG2D-L is immune stimulatory or inhibitory in the context of tumor and raised the concern as to whether soluble human NKG2D ligand is a cancer therapeutic target.

Harnessing NKG2D and the NKG2D ligand axis is an emerging avenue in cancer immune therapeutics [8,9,33,34]. To address the outstanding controversy and clear the pathway for therapeutic development to empower the NKG2D/NKG2D-L pathway for cancer immunotherapy, herein we utilized three mouse models to address the impact of the human sNKG2D-L, sMIC, on tumorigenesis and tumor progression. With three syngeneic tumor models that express human sMICB, we interrogated the differential impact of sMIC on tumor establishment and progression of established tumors with multiple disease models.

## 2. Material and Methods

### 2.1. Mice and Cell Lines

Cell lines of mouse melanoma B16F10, Lewis lung carcinoma LLC1, and the prostate tumor TRAMP-C2 were used in this study. All three cell lines were purchased from ATCC. The derivative cell lines, TRAMP-C2-sMICB, LLC-sMICB, and B16F10-sMICB cell lines, expressing the soluble human NKG2D ligand sMICB, were generated by transduction with an IRES-GFP retroviral vector containing the construct for recombinant soluble MICB, as described previously [35]. sMICB^+^ cells were selected by puromycin and further by flow cytometry sorting for GFP-positive cells and further validated by ELISA for sMICB in the culture supernatant (Appendix A). All cells were cultured in RPMI 1640 complete media with supplement of 10% FBS.

### 2.2. In Vivo Experiments

Mice were bred and housed under specific pathogen-free conditions in the animal facility at the Northwestern University in accordance with institutional guidelines for the approved Institutional Animal Care and Use Committee (IACUC) protocols. All mice used in this study were male rPB-MICB transgenic mice on the B6 background as previously described [36]. The rationale of using mice to be a sMIC-tumor host was previously described [36]. Briefly, the male rPB-MICB mice have the transgene of full-length human MICB in the prostate whose expression is controlled by the male hormone sensitive promoter rat probasin (rPB). As we have previously described, these mice have the identical phenotype as the wild-type B6 mice but are more tolerant to tumor cell lines expressing human sMIC, due to “endogenous” expression of MIC in the prostate [28,36]. To ensure that all tumor cells grew in the same background as the host, the rPB-MICB male mice were used for tumors. Tumor cells were implanted subcutaneously (s.c.) in the right flank region of cohorts of syngeneic rPB-MICB male mice with cell numbers indicated in each specific experiment. Mice were monitored for tumor incidence and tumor size once tumors were formed, three times per week. The tumor volume (calculated by L × W^2^/2) of 1800 mm^3^ was defined as the survival endpoint, unless otherwise specified.

### 2.3. Flow Cytometry Analysis

For analyzing tumoral NK and CD8 T cells, tumors were minced, gently meshed with syringe tips, and filtered through 70 μm mesh. Single-cell suspensions were first incubated for 20 min on ice with a viability dye and for 10 min with the anti-CD16/32 Fc-blocker solution, then stained with a combination of antibodies specific to cell surface markers for identification of lymphocyte subsets. These antibodies are: anti-CD45-AF700, anti-CD3-APC/Cy7 (clone 145-2C11), anti-NK1.1-PE (clone PK136), anti-CD8α-FITC (clone 53-6.7). To assess the capacity of IFN-γ production, tumor cell suspension was stimulated at 37 °C for 4 h with 50 ng/mL phorbol myristate acetate (PMA) and 500 ng/mL ionomycin with addition of 1 μM Golgi Plug during the last 2 h stimulation. Following stimulation, cells were stained with surface markers followed by fixation and permeabilization with BD Perm/Fix kits and an antibody specific to intracellular molecules IFN-γ. In some experiments, tumor cell suspension was incubated in complete RPMI 1640 media in the presence of 1 μM Golgi Plug for 2 h before being stained intracellularly for IFN-γ and granzyme B. All antibodies were mouse specific and were from BD Biosciences. Cells were analyzed using the BD Fortessa. Data were analyzed using the FlowJo v8 software (Tree Star).

For cell surface stem cell marker profiling, after incubation with the anti-CD16/32 Fc-blocker solution, tumor cells were stained with the following fluorochrome conjugated antibodies, respectively: anti-CD44-AF700 and anti-CD166-PE in combination with anti-SCA1-PE/Cy7 or anti-CD133-PE/Cy7. Each fluorochrome-conjugated isotype antibody was used as staining controls. PE-conjugated anti-mouse I-A/I-E antibody was used as a positive staining control and to examine tumor cell surface MHC I expression. For flow sorting to separate tumor cells with high and low stem cell markers, cells were stained with anti-CD44-AF700 and anti-CD166-PE for the separation of the CD44^Hi^CD166^Hi^ population from the CD44^Lo^CD166^Lo^ population. SCA-1 or CD133 expression of the sorted cell populations was examined by fluorochrome-conjugated respective antibodies.

### 2.4. Statistics

Wherever applicable, statistical data were expressed as mean ± standard error of the mean (SEM). Differences between means of populations were compared by an unpaired *t*-test. Tumor incidence was determined via a Kaplan–Meier analysis with “tumor incidence” as an occurring event. Mantel–Cox Log-rank test was used to analyze the significance level of tumor incidence among the two groups. A *p*-value of 0.05 or less was considered significant. GraphPad Prism software was used for all analyses.

## 3. Results

### 3.1. Model-Dependent Impact of Soluble Human NKG2D Ligand on Tumor Establishment

While most of the studies with human NKG2D ligands demonstrated that soluble NKG2D ligands suppress tumor immunity [29,37,38,39], controversy arose from findings that the overexpressing of soluble mouse NKG2D ligand, sMULT-1, in B16F10 tumor cells resulted in no tumor formation [32], suggesting that the role of soluble NKG2D ligands in regulating tumor immunity could be more complicated than what has been understood to date. To address the discrepancies between these study findings, we used three different syngeneic tumor models, the melanoma B16F10, the prostate tumor TRAMP-C2 (TC2), and the lung carcinoma LLC1, to compare tumor incidence between tumor cells engineered to overexpress the soluble human NKG2D ligand sMICB and the parental tumor cell lines. Secretion of sMICB was confirmed by ELISA in all three sMIC-overexpressing cell lines (Appendix A). Noteworthy, while mice do not express homologs of human NKG2D ligands MICA or MICB, human MICB or sMICB has been demonstrated to serve as a surrogate ligand for mouse NKG2D [30,31,35,36,40]. We subcutaneously implanted sMICB-expressing B16F10-sMICB, TC2-sMICB, and LLC1-sMICB and their parental counterparts B16F10, TC2, and LLC1 into syngeneic male mice that were engineered to express MICB specifically in the prostate under the androgen-sensitive promoter rat probasin (rPB, hence referred as rPB-MICB mice). Tumor incidence, time to tumor formation, and the growth dynamics of established tumors were monitored (Figure 1). The rationale for using rPB-MICB male mice as the host rather than wild type B6 mice for sMICB-expressing tumors was described previously [28,36]. To briefly reiterate, the rPB-MICB mice develop less immunity against implanted tumor cells overexpressing human MIC due to “endogenous” expression of MICB in the prostate upon puberty.

The B16F10-sMIC tumors demonstrated a significant delay in tumor establishment as compared to the parental B16F10 tumors (Figure 1A). On average, B16F10-sMICB tumors arose on day 23 post-inoculation, whereas the parental B16F10 tumors arose on average on day 6 post tumor inoculation (*p* < 0.05, Figure 1A). By day 26 post-tumor injection, tumors were established in 50% of mice inoculated with B16F10-sMICB tumors. These observations demonstrated that overexpression of the soluble human NKG2D ligand sMICB in B16F10 tumors resulted in a delay in tumor establishment, not the complete tumor rejection as being reported [32].

In contrast to the B16F10-sMICB melanoma model, the prostate TRAMP-C2-sMICB (TC2-sMICB) cells presented a significantly more rapid tumor onset than the parental TRAMP-C2 (TC2) cells after inoculation (*p* < 0.05, Figure 1B). Specifically, TC2-sMICB tumors arose as early as day 7 post inoculation and reached 100% tumor penetration by day 11, whereas parental TC2 tumor onset did not occur until day 11 and did not reach 100% tumor penetration until day 22 post inoculation (*p* < 0.05, Figure 1B). Interestingly, different from the B16F10-sMICB or the TC2-sMICB models, no significant difference in tumor establishment or tumor onset was observed between LLC-sMICB and the parental LLC tumors. LLC-sMICB and the parental LLC tumors displayed a similar timeline in tumor establishment, both of which reached 100% tumor penetration within 12 days post tumor inoculation (Figure 1C). Together, these data demonstrated a model-dependent effect of sMICB on tumorigenicity or tumor initiation.

### 3.2. Model-Independent Impact of sMIC on the Growth of Established Tumors and on the Function of Tumoral Effector Cells

Irrespective of tumor incidence or the time of tumor onset, once tumors were initiated or formed, in all three disease models, sMIC-overexpressing tumors grew at a significantly more aggressive rate than the parental sMIC-negative tumors (Figure 2). The accelerated tumor growth with sMIC overexpression is in agreement with previous studies by multiple investigators in various experimental settings demonstrating that tumor produced sMIC can significantly comprise NK and CD8 T cell function [16,20,22,24,26,27,30,31,36,41]. To validate that the more aggressive tumor growth of sMICB-expressing tumors in current experimental settings is associated with impairment of NK and CD8 T cell immunity, we assayed the function of tumor-infiltrated NK and CD8 T cells (Appendix A), both of which express the receptor NKG2D for sMICB and are critical anti-tumor effector cells. In all three models, there was a significant reduction in NK cell content in tumor infiltrates when tumor cells expressed sMICB (Figure 3A, *p* < 0.05 for all models), which is consistent with the previous finding in the TRAMP/MICB autochthonous tumor model that tumor-produced sMIC impairs the NK cell’s homeostatic self-renewal ability [36]. NK cells in sMICB-expressing tumors had a significantly reduced response to PMA/I stimulation as measured by IFNγ expression (Figure 3B, *p* < 0.05 for all models), one of the key effector molecules for NK cell mediating anti-tumor immunity. sMIC expression also significantly impaired tumoral CD8 T cell response to PMA/I stimulation as measured by IFNγ expression (Figure 4A, *p* < 0.05 in all models); however, CD8 T cell content in tumor infiltrates was not significantly impacted by sMICB (Figure 4B). The impaired response to external PMA/I stimulation ultimately signifies the comprised ability of these cells to respond to tumor-specific stimulations in vivo. To support this concept, we conducted representative analyses of TRAMP-C2 and TRAMP-C2-sMICB tumoral NK and CD8 T cell IFNγ and granzyme B expression in an ex vivo setting without PMA/I stimulation. Corroboratively, the ability of tumoral NK and CD8 T cells to produce IFNγ and granzyme B without PMA/I stimulation was shown to be significantly compromised in TRAMP-C2-sMICB tumors as compared to TRAMP-C2 tumors (Appendix A). Collectively, these data re-reinforce the concept that soluble human NKG2D ligands negatively regulate anti-tumor NK and CD8 T immunity in established tumors with more profound impact on NK cells.

### 3.3. Tumor Cell Stem-like Property Rather Than sMIC Expression Determines the Ability of In Vivo Tumor Establishment

Our data in three different models demonstrated that sMIC facilitates growth or progression once tumors are formed; however, the impact of sMIC on the ability of tumor formation or time required for tumor formation appeared to be model-dependent. We further sought to understand the discretionary impact of sMIC on established tumor growth versus the potential of tumor formation. Given that stem-like features could profoundly enhance the ability of tumor cells to colonize and form tumors in vivo [42,43,44], we thus examined the common cancer stem cell surface markers in the three sMIC-expressing cell lines and their parental cell lines with a flow cytometry assay. Noteworthy, different tumor types possess different stem cell makers [45]; we thus selected the most common markers, CD44, SCA-1, CD166, and CD133 [45,46,47], in our analyses. Interestingly, there is a general correlation of expression of stem cell markers with the ability of tumor formation or the time required for tumor onset, irrespective of sMIC expression (Figure 5). Specifically, TC2-sMICB cells expressed higher levels of CD166 than the parental TC2 cells (Figure 5A), corresponding with higher tumorigenicity of TC2-sMICB cells than TC2 cells. There was no difference in the expression of stem cell markers between LLC-sMICB cells and the parental LLC cells (Figure 5B), correlating with no significant differences in the incidence of tumor formation or the time required for tumor formation between the two cell lines. Surprisingly, the parental B16F10 cells express high levels of stem cell markers, CD44, CD166, whereas the B16F10-sMICB cells presented a heterogenic population in the expression of CD44 and CD166 (Figure 5C), which potentially reflected the heterogenic tumor incidence in vivo as presented in Figure 1.

To further confirm that the ability of tumor formation of the sMIC-expressing tumor cell was determined by tumor cell stem-like property rather than sMIC-associated immune editing, we separated the CD166^Hi^CD44^Hi^ B16F10-sMICB cell population (termed Hi-stem marker) from the CD166^Lo^CD44^Lo^ cell population (termed Lo-stem marker) of B16F10-sMICB cells by flow cytometry sorting (Figure 6A). Interestingly, the sorted CD166^Hi^CD44^Hi^ B16F10-sMICB cells also express a higher level of SCA-1 as compared to the sorted CD166^Lo^CD44^Lo^ B16F10-sMICB cells (Figure 6A). Both populations express either no or an extremely low level of CD133 (Figure 6A). After sorting, we further confirmed that both cell lines secreted a comparable amount of sMICB in the culture (Appendix A). We inoculated the isolated B16F10-sMICB High-stem marker (CD166^Hi^CD44^Hi^) and the B16F10-sMICB Low-stem marker cells (CD166^Lo^CD44^Lo)^, respectively into rPB-MICB mice. We inoculated the parental B10F10 (Hi-stem cell mark) CD44^Hi^CD166^Hi^ into rPB-MICB mice as the comparison. As shown in Figure 6B, at both cell doses, 100,000 cells/mouse and 300,000 cells/mouse, the Lo-stem marker B16-sMICB cells presented a much lower tumor formation rate as compared to the Hi-stem marker B16-sMICB cells (*p* < 0.001). These data suggest that tumor cell stem-like property rather than sMIC-mediated immune editing determines the potential of tumor formation of sMIC-expressing tumor cells.

## 4. Discussion

The objective of this study is to understand and to resolve the discrepancies or controversies in the literature with regard to whether soluble NKG2D ligands are immunosuppressive or immunostimulatory in anti-tumor responses. Utilizing three different animal models expressing the soluble human NKG2D ligand sMICB, we clearly distinguished the differential impact of soluble human NKG2D ligands on the potential of initial tumor formation versus the progression of established tumors. Our data demonstrated that the temporal requirement for tumor establishment was dominantly determined by tumor cell stem-like features, irrespective of sMICB expression; however, in established tumors, soluble NKG2D ligands facilitated more aggressive growth and induced the impaired function of NK and CD8 T cells irrespective of tumor models. Thus, the discrepancies in the literature are largely context-dependent, whether the question was set to address tumor formation or incidence versus growth rate of established tumors.

We demonstrate that the impact of sMICB on tumor formation was highly model-dependent. Tumor cell characterization showed that the temporal-dependent tumor establishment or ability to form tumors by sMICB-expressing tumor cells (as compared with their respective parental tumor cells) was strongly associated with the tumor cell expression of stem cell-like markers, represented by the levels of surface CD44 and CD166 expression. In both TRAMP-C2 and LLC tumor models, the correlation of the ability of tumor formation and cancer cell stem-like feature was evident, irrespective of sMICB expression. In the melanoma model, we showed a heterogenic pattern of tumor formation with B16F10-sMICB tumor cells as compared to the parental B16F10 tumor cells. The heterogeneric pattern of tumor formation was associated with the heterogenic expression level of tumor stem cell markers. With the separation of B16F10-sMICB cells expressing high stem-like markers, such as CD44 and CD166, and B16F10-sMICB cells with low or no expression of these stem-like cell markers, we demonstrated that B16F10-sMICB cells expressing high stem-cell markers had the comparable ability to form tumors with the parental B16F10 cells; whereas B16F10-sMICB cells with low or no expression of cell surface stem-like markers had a significantly impaired ability to form tumors, presented as a failure to form tumors or a delay in tumor formation, as compared to B16F10 tumor cells or B16F10-sMICB cells expressing high levels of stem-like markers.

It is noteworthy that CD166 is a recently described cancer stem marker that has been shown to dictate tumor cell self-renewal independent of CD133. Expression of CD166 can control tumor cell migration and metastasis and is correlated with poor disease outcome in a range of cancer types [48,49,50,51]. It was shown that knockdown of CD166 in tumor cells caused a delay in myeloma arising in mice [52], confirming the significance of CD166 in controlling tumor establishment.

It is unclear in this study how expression of sMICB modulates the B16F10 cell stem-like feature to differentiate a subset of cell populations into a less stem-like state. Studies have shown that certain tumor types can express NKG2D and that autonomous NKG2D signaling in tumor cells can modulate their plasticity [53,54,55]. It was demonstrated that NKG2D^+^ ovarian cancer cell populations harbor substantially greater capacities for self-renewing in vitro and in vivo tumor initiation in immunodeficient NSG mice than NKG2D^-^ cell population controls. However, we did not detect NKG2D expression on B16F10 tumor cells, suggesting that this is not the potential pathway.

Our data presented a discrepancy with the study by Deng et al. in that expression of the mouse soluble NKG2D ligand, sMULT-1, inhibited tumor establishment [32]. There are several experimental differences between our study and the study by Deng et al. First, in the study by Deng et. al, tumor incidence was followed only up to 19 days post-tumor inoculation; the outcome with a longer-period of follow-up was unclear. We have followed the tumor incidence for a much longer period of time. Second, B16F10 was the only model being used in the study by Deng et al., and whether sMULT-1 would have the same impact in models beyond B16F10 was unknown [32]. Lastly, it is also possible that the human NKG2D ligand and the mouse NKG2D ligand act differently due to different affinity in binding to NKG2D. Nonetheless, our studies presented a model-dependent impact of human soluble NKG2D ligand on tumor establishment via impact on tumor cell stem-like property.

Consistent with the concept that a soluble NKG2D ligand was immunosuppressive, we show that once a tumor is established, sMIC-expression facilitated a more aggressive tumor growth in all three tumor models. Corroboratively, this study demonstrated that tumoral content of NK cells and the ability of tumoral NK and CD8 T cells to produce IFNγ were significantly reduced in sMICB-expressing tumors. In recent studies of multiple pre-clinical therapeutic settings, using antibodies to block human MIC shedding or to clear sMIC demonstrated a benefit in reducing tumoral sMIC and enhancing anti-tumor immunity [26,27]. Prior to these studies, it was demonstrated in the bi-transgenic TRAMP/MICB mouse model that an antibody-clearing tumor-shed, sMIC, resulted in de-bulking of tumors and eliminating metastasis through reinvigorating NK and CD8 T cell function [29,30,31,36]. It was also reported that patients with high levels of circulating sMIC presented a poor clinical response to immune checkpoint inhibitor therapy [24,41]. In retrospect, before the era of immunotherapy, clinical studies by Jinushi et al. demonstrated that cancer patients who developed anti-sMICA antibodies to clear sMIC during treatment with cancer vaccines or anti-CTLA-4 treatment, had better clinical responses and enhanced NK cell anti-tumor activity [21,56]. Collectively, this study reiterates the immunosuppressive nature of soluble human NKG2D ligands in suppressing anti-tumor immunity in established tumors.

NK cell activation is controlled by the amplified signals from its activating receptors over its inhibitory receptors. Sufficient and proper ligand stimulation of a given NK cell activating receptor can lead to acute NK cell activation [57,58]. However, prolonged stimulation of NK activating receptors could lead to NK cell desensitization or functional polarization. It was reported that chronic NKG2D stimulation by its ligand can lead to impaired NK cell function in both murine and human systems [59,60]. Uncontrolled or chronic NKG2D stimulation by its soluble ligands can skew NK cell to a proinflammatory phenotype with no or limited cytotoxicity [40,61]. Together, these studies corroboratively support the biology that consistent stimulation with sMIC produced by established tumor cells will lead to impaired NK and CD8 T cell function and thus allow tumors to grow more aggressively.

In conclusion, we differentiated the impact of soluble NKG2D on controlling tumor initiation versus progression of established tumors. We clearly demonstrated that soluble NKG2D ligands had no or little impact on tumor initiation and that tumor cell stem-like property dictated the time of tumor initiation in vivo. Our data further underscores the impact of soluble NKG2D ligands, at least human soluble NKG2D ligand sMIC, on the progression of established tumors. Importantly, our study clarified the discrepancies or controversies regarding whether soluble NKG2D ligands suppress or stimulate anti-tumor immunity. Given that sMIC was mostly found elevated in patients with metastatic diseases and associated with poor survival in a broad range of malignancies [62,63,64] and that sMICB is significantly elevated in patients with metastasis and poor response to immune checkpoint blockade therapy [24,41,64], our study reinforced the translational potential of targeting the soluble NKG2D ligand, sMIC, for metastatic diseases.

## Figures and Tables

**Figure 1 biomedicines-12-00196-f001:**
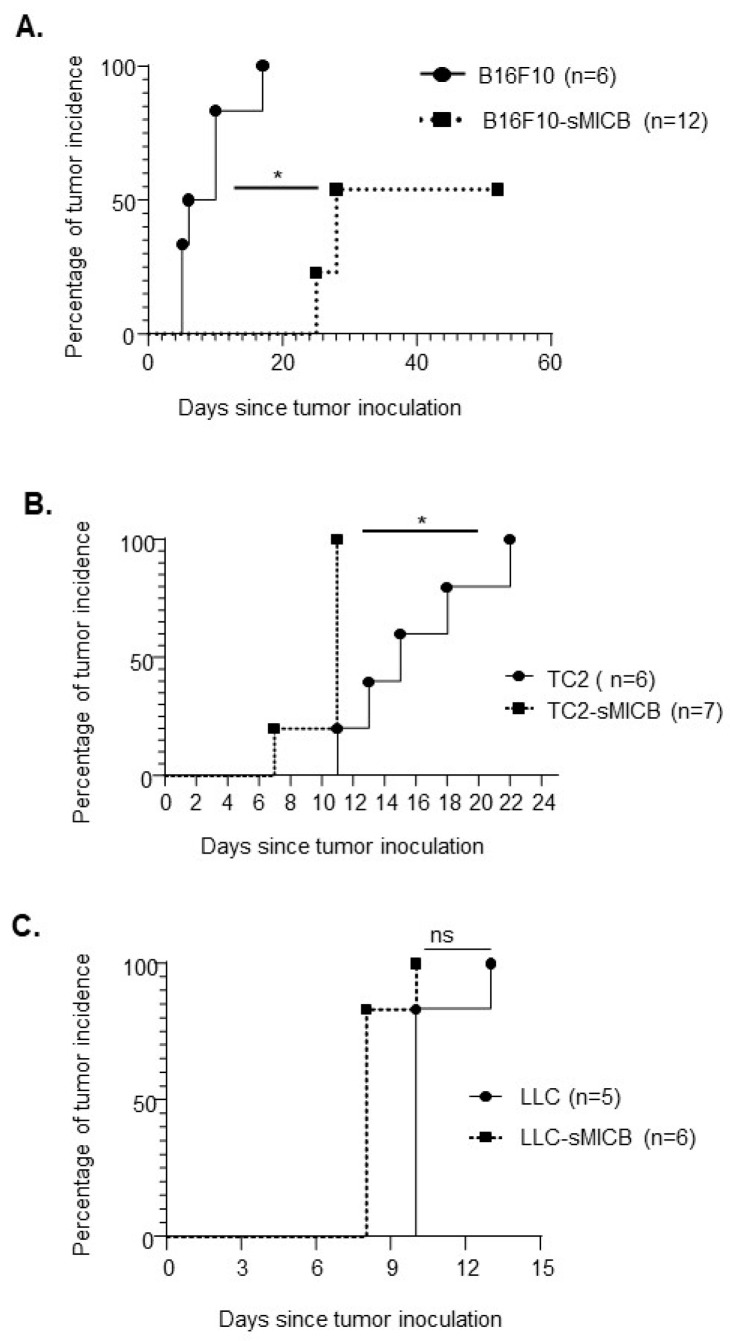
Overexpression of the soluble human NKG2D ligand sMICB differentially impacts tumor incidence and the time of tumor onset in different tumor models. Comparison of tumor incidence between parental tumors and sMICB overexpressing tumors. The percentage of mice that remained tumor free over time was compared. Tumors were subcutaneously inoculated with three disease models: (**A**) melanoma model B16F10 vs. B16F10-sMICB with 5 × 10^5^ cells/mouse being inoculated; (**B**) prostate tumor model TRAMP-C2 (TC2) vs. TRAMP-C2-sMICB (TC2-sMICB) with 1 × 10^6^ cells/mouse being inoculated; and (**C**) lung tumor model LLC vs. LLC-sMICB with 5 × 10^5^ cells/mouse being inoculated. Note that the number of cells being inoculated with each model was based on the published literature. Tumor incidence was accounted when the tumors were palpable. Tumor incidence was determined via Kaplan–Meier analysis with “tumor incidence” as an occurring event. The Mantel-Cox Log-rank test was used to analyze the significance level of tumor incidence among two groups (ns, not significant). * *p* < 0.05.

**Figure 2 biomedicines-12-00196-f002:**
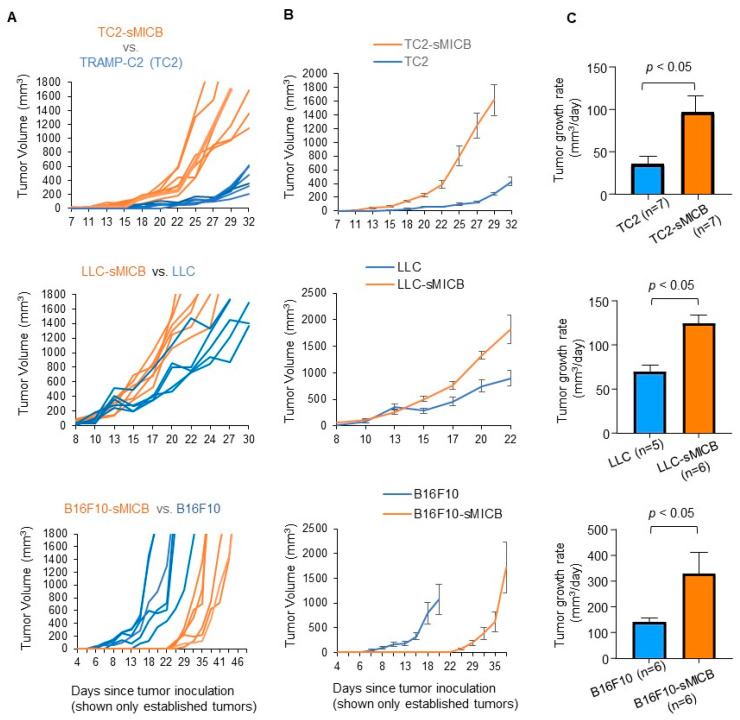
sMICB facilitates more aggressive growth of established tumors in all disease models. (**A**) Individual tumor growth curves (**A**) in three tumor models of parental tumor cells and sMICB-overexpressing tumor cells. Each line represents one individual mouse. (**B**) Average tumor growth curve in mice where a tumor arose. Of note, due to tumors in individual animals reaching the IACUC approved maximum size at very early time points, the average tumor growth data presented in (**B**) is before any given animal reached the maximum tumor volume and thus presented a shorter time period than individual mouse tumor growth curves. (**C**) Comparison of established tumor growth rate between parental tumor cell line and corresponding sMICB-expressing cell lines. Tumor growth rate was calculated by linear regression using Prism software. Statistical significance in growth rate comparison was determined by unpaired *t*-test. Noteworthy, only established tumors were taken into consideration for growth curves and growth rates.

**Figure 3 biomedicines-12-00196-f003:**
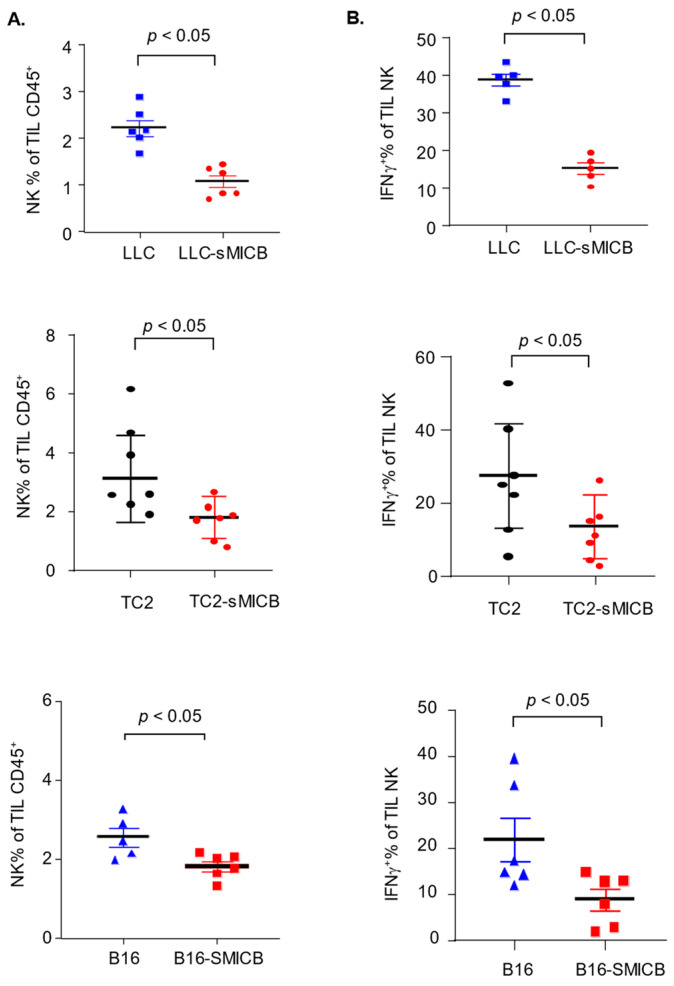
Impaired NK number and function in established sMIC-positive tumors across models. (**A**) Significantly reduced tumoral NK cell content in all sMIC-expressing tumor models as compared to their parental lines (*p* < 0.05). (**B**) Significantly (*p* < 0.05) reduced ability of IFNγ production of tumoral NK cells from sMICB-expressing tumors as compared to parental sMIC-negative tumors. Statistical significance was determined by unpaired *t*-test.

**Figure 4 biomedicines-12-00196-f004:**
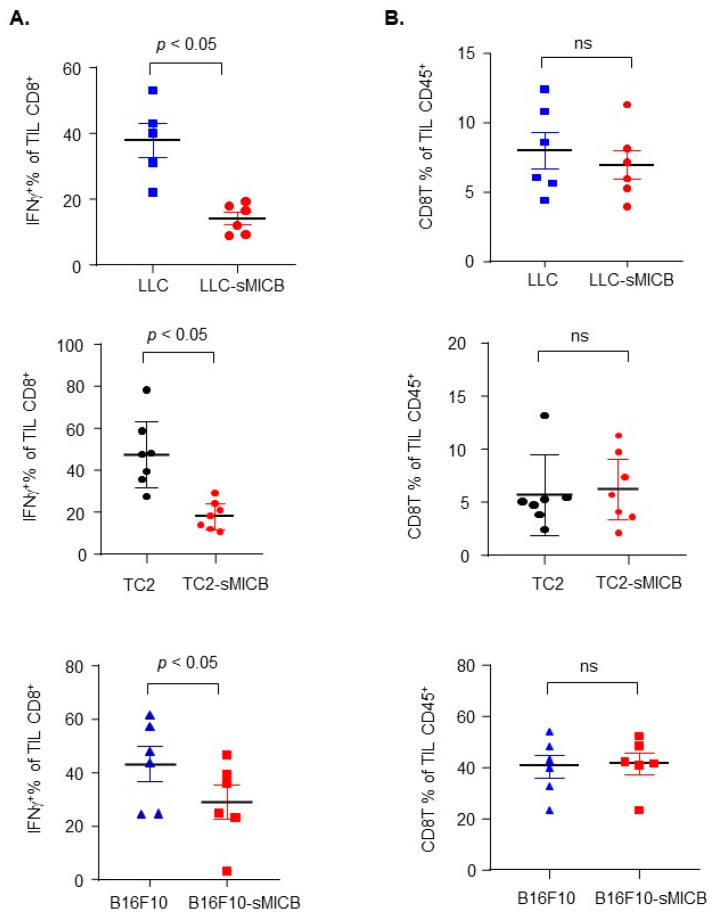
Impaired CD8 T cell function but not CD8 T cell content in established sMIC-positive tumors across models. (**A**) Significantly (*p* < 0.05) reduced ability of IFNγ production of tumoral CD8 T cells from sMICB-expressing tumors as compared to parental sMIC-negative tumors. (**B**) No significant difference in tumor CD8 T cell content between sMIC-expressing tumor models as compared to their parental models (*p* < 0.05), ns: no significance. Statistical significance was determined by unpaired *t*-test.

**Figure 5 biomedicines-12-00196-f005:**
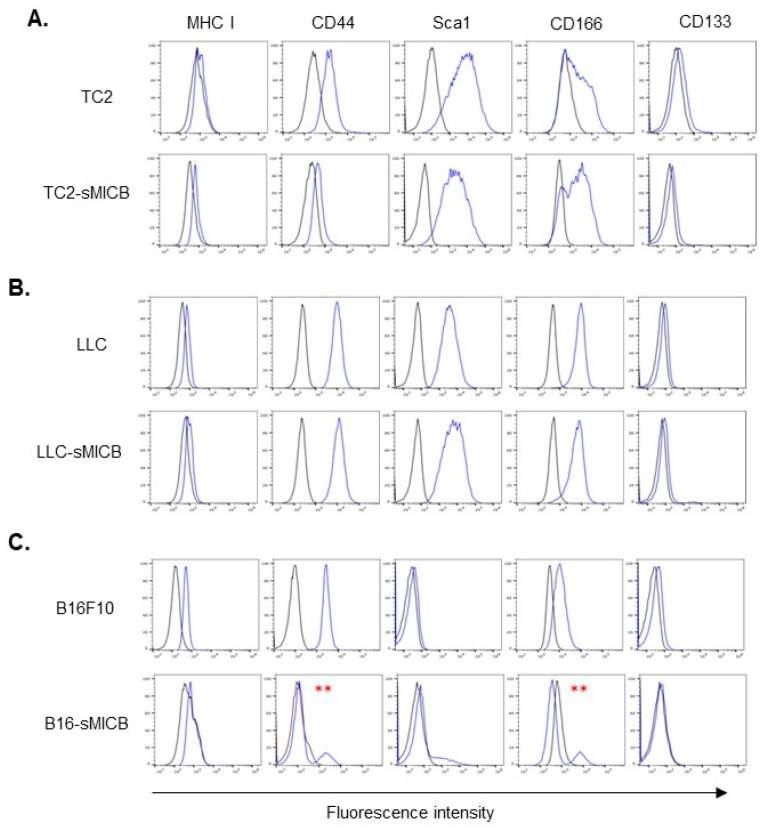
Stem cell marker comparison of TRAMP-C2 (TC2)-sMICB (**A**), LLC-sMICB (**B**), and B16-sMICB (**C**) with respective parental counterparts. The established mouse tumor cell stem cell markers SCA1, CD166, and CD133 were characterized by flow cytometry assay. Black profile represents control IgG staining. Blue profile represents specific marker staining. Surface MHCI was also characterized as a positive staining control. MFI, Mean Fluorescence Intensity. ** indicates differences in expression levels with the respective comparing counterpart.

**Figure 6 biomedicines-12-00196-f006:**
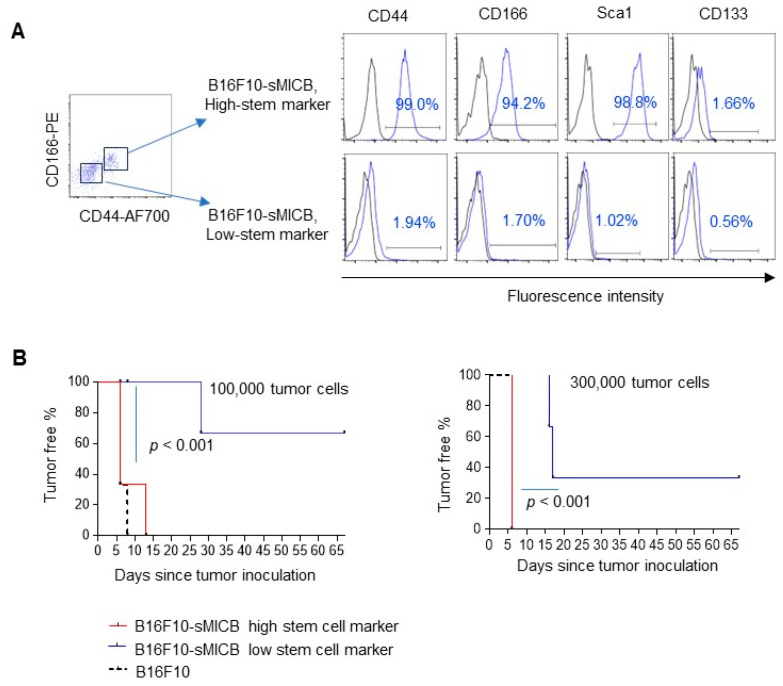
Tumor cell stemness impacts the tumorigenic potential of B16-sMICB tumors. (**A**) Flow-cytometry sorted B16-sMICB cells with low expression of surface stem cell markers (CD44^Lo^CD166^Lo^) and high expression of surface stem cell markers (CD44^Hi^CD166^Hi^). (**B**) Tumorigenic ability of B16F10-sMICB tumors with Hi-stem cell markers and Lo-stem cell markers that are inoculated with 300,000 cells and 100,000 cells, respectively. N = 6, six mice per group. Tumor incidence was determined via Kaplan–Meier analysis with “tumor incidence” as an occurring event. Mantel-Cox Log-rank test was used to analyze the significance level of tumor incidence among two groups.

## Data Availability

Research data can be shared upon request.

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
