# Peer review of "The Human Soluble NKG2D Ligand Differentially Impacts Tumorigenicity and Progression in Temporal and Model-Dependent Modes"

_biomedicines, 2024, doi:10.3390/biomedicines12010196_

Round 1

Reviewer 1 Report

Comments and Suggestions for Authors

This is a potentially very interesting study that examines the relationship between soluble ligands and tumorigenicity and T/NK cell immune function. The authors have also uncovered an interesting relationship between the expression of soluble ligands and stem cell-like markers with tumor immunogenicity.

Major points of concern:

1.     It would strengthen the authors message if the authors could provide data to prove that the B16F10, LLC1, and TRAMP2 tumor cell-lines actively secrete the soluble MICB from the IRES-GFP construct that they have been transfected with since GFP expression alone does not necessarily prove that the sMICB is expressed in these lines.

2.     Figure 2: Perhos I’m wrong but I don’t understand the title ‘in all disease’ models since B16F10 model is clearly the reverse of the TRAMP-C2 and LLC models?

3.     Wouldn’t you expect more IFN-g+ NK and T cells in the B16 models if sMICB leads to lower tumor volume?

Minor points:

1.     The references are heavily focused on NKG2D from a restricted set of labs and do not consider the broader literature from other groups that shows secreted ligands can directly stimulate activating NK cell receptors (PMID: 29275861).

Author Response

We sincerely thank you for recognizing the significance of the study and for the constructive critiques. We have revised the manuscript accordingly and addressed all critiques point-by-point below.

Major Critique 1:

Response: We have now included ELISA data in Supplement Figure S1 demonstrating that all three sMIC-expressing cell lines secrete sMIC.

Major Critique 2:

We apologize for the confusion. We would like clarify that the title of Figure 2 is “sMICB facilitates more aggressive growth of established tumors in all disease models”. While it is true that B16F10 model is the reverse in tumor formation as presented in Figure 1; however, once tumor is formed,  all sMIC-expressing tumor grew significantly faster than the parental tumor cell lines. This is shown in the Figure 2c. To avoid the confusion, we have revised the legend of Figure 2c as “Comparison of established tumor growth rate between parental tumor cell line and corresponding sMICB-expressing cell lines”. 

Major Critique 3:

Yes. Indeed in established tumors as we have shown in Figure 2c.

Minor Critique 1:

Thank you for bringing our attention to this important study. We have include this article in our discussion (paragraph 7, marked).

Response to Review #2.

General response:

Thank you for the input and questions. We would like to emphasize the following points in order to better address the critique:

  1. The main objective is this study is to distinguish the impact of soluble NKG2D ligand on tumor establishment vs. tumor progression of established tumors to resolve the outstanding controversy in the literature.

In the literature, the impact of soluble NKG2D on tumor immunity is controversy. In a large body of studies, human soluble NKG2D ligand has been shown to be immune suppressive, in suppressing both NK and CD8 T cell function, in various settings. There are more than 50 research articles in PubMed consistently reporting how soluble NKG2D impacted NK and CD8 T cells function. There are also a large body of publications describing the underlying mechanisms. However, these largely body of consistent study findings were challenged by one study published in Science in 2015 (Deng et al., PMID25745066). In this study, a soluble mouse soluble NKG2D ligand sMULT-1 was overexpressed in B16 tumor cell lines. Tumor incidence was compared between parental B16 and sMULT1-B16 cell lines at a limited time frame that is 19 days post tumor inoculation. It was found that no tumors were established at Day 19 in mice inoculated with sMULT1-B16 cells, while all mice inoculated with parental B16 cells had tumor established. Without follow-up longer period of study or 2nd tumor model, the authors concluded that ‘tumor shed soluble NKG2D ligand stimulated tumor immunity”. To our view, the study lacks rigor to draw such a conclusion. Here, we conducted a similar study. Our initial observation of tumor incidence between B16-sMICB and B16 cell is consistent with the study observed by Deng et al in a limited observation period. However, with prolonged period of observations, the conclusion by Deng et al is not supported experimental data. More experimental models of LLC and TRAMP-C2  further challenges the conclusion by Deng et al. Here we presented a study that were able to reproduce the data reported by Deng et al of B16 tumor model of tumor incidence in a limited time frame; further beyond, we provoked the findings by Deng et al., by longer period of follow-up study and multiple tumor models. The purpose of the study is not intended to demonstrate how souble NKG2D ligand acts on NK and CD8 T cells function. As mentioned earlier, the impact of soluble NKG2D ligand on NK and CD8 T cell function has been well demonstrated in vitro and in vivo by many published studies. The goal of this study is not to duplicate these published studies.

We have included all relevant literature in the revised manuscript.

Response to Major concern 1

  1. Thank you for the input. In our results, we have provided data of IFNg expression of tumor infiltrated NK and CD8 T cells in their ability in response to PMA/I stimulation, which is a simple and ultimate 1st step to test the ability or machinery of NK cell and CD8 T cell response to a generic stimulation. If NK and CD8 T cells have impaired ability to respond to MA/I stimulation, it is unlikely that they will respond well to tumor specific stimuli. In the revised manuscript, we provided representative and corroborative data of tumors NK and CD8 T cells IFNg and granzyme expression in TRAMP-C2 and TRAMP-C2-sMICB tumors without PMA/I stimulation (Figure S3). Because of these rationales, we did not further analyze tumoral NK and CD8 T cell function in B16 and LLC model without PMA/I stimulation. We did not view these elaborative experiments will provide us more insights into the biology.

  1. Technically, there are very few NK cells in tumors, although very powerful. Thus it is a rather challenging task to isolate tumor infiltrated NK cells for in vitro cytotoxicity assay. Although there are a lot more CD8 T cells in the tumor infiltrated, but those are heterogenic CD8 T cells. In vitro cytotoxicity assay will not be antigen-specific, but limited to CD3/CD28 stimulation, which will not provide more information than what we have presented.

Reviewer 2 Report

Comments and Suggestions for Authors

Serritella et al. undertook very interesting problem of differential outcome NK cell receptor ligands on NK cells. NKG2D, which is an activating receptor and which downregulation was associated with inhibition of NK cell anti-viral and anti-tumor activity, may, in fact, play contrary role in tumor progression. Thus, as the background and the concept of the manuscript is of great importance for researchers studying NK cell biology. Nonetheless, presented results seem very preliminary and the manuscript needs improvement in order to be published.

Main issues:

1.      The results lacks any NK cell/CD8 cell functional analysis. There is data on IFNgamma expression in the cells isolated from studied tumors, yet it too little to know what exactly is happening with the lymphocytes stimulated with soluble NKG2D ligands. It would be good to add cytotoxicity assay results of NK cells, granzyme B/perforin expression levels, TNF production, activation/exhaustion markers expression on CD8 cells etc.

2.      The question if and why NKG2D ligands inhibit NK/CD8 cells was not answer – it would be beneficial to check what is the effect of high concentrations of NKG2D ligands on NK cells antitumor activity, even in in vitro model.

3.      I do not understand what is the association of tumor stem cell markers and the aim of the study.

Minor issues:

1.      NK cell and CD8 cell phenotyping plots should be added.

2.      Statistical tests applied should be stated in every figure description.

3.      Figure 2 – comparisons between bars are not centered.

4.    CD8 T cell content data should be presented on the figure 4, even if there are no significant differences.

5.      Number of mistakes such as double “.”, extra enters etc. was noticed – the whole manuscript should be then thoroughly checked.

Author Response

Response to Review #2.

We sincerely thank you for recognizing the significance of the study and for the constructive critiques. We have revised the manuscript accordingly and addressed all critiques point-by-point below.

We would like to provide more background of this  study here before our point-to-point response:

The main objective is this study is to distinguish the impact of soluble NKG2D ligand on tumor establishment vs. tumor progression of established tumors to resolve the outstanding controversy in the literature.

In the literature, the impact of soluble NKG2D on tumor immunity is controversy. In a large body of studies, human soluble NKG2D ligand has been shown to be immune suppressive, in suppressing both NK and CD8 T cell function, in various settings. There are more than 50 research articles in PubMed consistently reporting how soluble NKG2D impacted NK and CD8 T cells function. There are also a large body of publications describing the underlying mechanisms. However, these largely body of consistent study findings were challenged by one study published in Science in 2015 (Deng et al., PMID25745066). In this study, a soluble mouse soluble NKG2D ligand sMULT-1 was overexpressed in B16 tumor cell lines. Tumor incidence was compared between parental B16 and sMULT1-B16 cell lines at a limited time frame that is 19 days post tumor inoculation. It was found that no tumors were established at Day 19 in mice inoculated with sMULT1-B16 cells, while all mice inoculated with parental B16 cells had tumor established. Without follow-up longer period of study or 2nd tumor model, the authors concluded that ‘tumor shed soluble NKG2D ligand stimulated tumor immunity”. To our view, the study lacks rigor to draw such a conclusion. Here, we conducted a similar study. Our initial observation of tumor incidence between B16-sMICB and B16 cell is consistent with the study observed by Deng et al in a limited observation period. However, with prolonged period of observations, the conclusion by Deng et al is not supported experimental data. More experimental models of LLC and TRAMP-C2  further challenges the conclusion by Deng et al. Here we presented a study that were able to reproduce the data reported by Deng et al of B16 tumor model of tumor incidence in a limited time frame; further beyond, we provoked the findings by Deng et al., by longer period of follow-up study and multiple tumor models. The purpose of the study is not intended to demonstrate how souble NKG2D ligand acts on NK and CD8 T cells function. As mentioned earlier, the impact of soluble NKG2D ligand on NK and CD8 T cell function has been well demonstrated in vitro and in vivo by many published studies. The goal of this study is not to duplicate these published studies. We have included all relevant literature in the revised manuscript.

Response to Major concern 1

  1. Thank you for the input. In our results, we have provided data of IFNg expression of tumor-infiltrated NK and CD8 T cells in their ability of responding to PMA/I stimulation, which is a simple and ultimate 1st step to test the ability or machinery of NK cell and CD8 T cell response to a generic stimulation. If NK and CD8 T cells have impaired ability to respond to MA/I stimulation, it is unlikely that they will respond well to tumor-specific stimuli. In the revised manuscript, we provided representative and corroborative data of tumors NK and CD8 T cells IFNg and granzyme expression in TRAMP-C2 and TRAMP-C2-sMICB tumors without PMA/I stimulation (Figure S3). Because of these rationales, we did not further analyze tumoral NK and CD8 T cell function in B16 and LLC models without PMA/I stimulation. We did not view these elaborative experiments will provide us more insights into the biology.

  1. Technically, there are very few NK cells in tumors, although very powerful. Thus it is a rather challenging task to isolate tumor infiltrated NK cells for in vitro cytotoxicity assay. Although there are a lot more CD8 T cells in the tumor infiltrated, but those are heterogenic CD8 T cells. In vitro cytotoxicity assay will not be antigen-specific, but limited to CD3/CD28 stimulation, which will not provide more information than what we have presented.

Response to Major concern is Review 2.

There is a large body of literature describing how NKG2D ligands impair NK/CD8 T cell function. These studies have been cited in our manuscript (ref 16,20,22,24,26,27,30,31,38,41). Importantly, our goal of the current study is NOT to provide further understanding of how NKG2D ligands impair NK/CD8 T cell function.

Response to Major issue 3 of Reviewer 2:

The Aim of the goal is to distinguish the impact of soluble NKG2D ligands on tumor establishment vs. tumor progression of established tumors. We found that tumor initiation or establishment was not determined by soluble NKG2D ligands as demonstrated by Deng et al; instead, it was associated with tumor cell stemness. We hope that this clarifies the major concern. We acknowledge the limitation of the current study that we did not address how soluble NKG2D ligand expression changes tumor cell stemness features. This warrants further investigations.

Response to Minor issues of Review 2:

  1. We have added NK and CD8 T cell phenotyping plots in Supplement Figure S2.
  2. We have added the statement of statistical tests in every figure description.
  3. We have centered the placement of all the comparison bars.
  4. We have the original data of CD8 T cell content in Figure 4B.
  5. We have manually proof-read the entire manuscript and corrected all errors. Thank you for pointing this out.

Reviewer 3 Report

Comments and Suggestions for Authors

1.       Can the authors elaborate why there is no immune response against the human soluble MICB in mice? The authors provide 2 references (both from their previous publications) but none of them provide the rationale why using a rat promoter to drive the ectopic expression of human sMICB could bypass mouse immune recognition. It does not make any sense to me. Please elaborate on this.

2.       In figure 1, where are the tumor cells injected into? The melanoma cells might be fine at any injection sites subcutaneously, while lung cancer and prostate cancer cells might not be in the appropriate microenvironment which is reflected by the two opposite outcomes between melanoma and lung/prostate model. Therefore, it is unknown whether this is a model dependent outcome or just a questionnaire of incorrect tumor injection site/grow microenvironment.

3.       In figure 2, please provide the rationale to remove samples for statistical calculations. Ex, figure 2 top panel, there are 7 samples in the panel A but 6 samples in panel C. In the mid panel, there are 6 samples for both groups in panel A and 5 samples in panel C. In the lower panel, there are 6 samples for both groups in panel A but surprisingly 3 samples are removed for calculation in panel C for sMICB. Why are these samples removed? For panel A (lower panel), is there one mouse which never develops tumor (the flat orange line for sMICB)?

4.       In figure 2, how is tumor growth rate calculated? For B16F10 model, the sMICB overexpression had delayed tumor development but the tumor growth rate was more than double compared to the control.

5.       Please include the methods/materials of stem cell marker analyses in the manuscript

6.       For figure 3 and 4, why PMA and ionomycin were added to stimulate cytokine production? Shouldn’t it be more reasonable to study the natural NK and CD8 T cell functionality in tumor overexpressing the sMICB ? There might be no difference at all naturally but the exogenous stimulation with PMA/Ionomycin might manipulate the immune cell function and lead to false positive signals.

7.       The resolution of flow images needs to be improved.

8.       To clarify the stem cell sorting, are SCA1, CD166 and CD44 (presented in the result section), SCA1 and CD44 (presented in figure 6A), or the SCA1 and CD166 (presented in the figure 3 legend) used as parameters for sorting? As the authors have 3 completely different statements about stem cell sorting, please clearly define which statement is correct.

9.       Line 49; page 2: overexpression

10.   Please improve the writing as some sentences do not make sense at all.

Comments on the Quality of English Language

See comments above

Author Response

We sincerely thank the reviewers for recognizing the significance of the study and for the constructive critiques. We have revised the manuscript accordingly and addressed all critiques point-by-point below.

Response to Comment 1:

We apologize that we did not explicitly explain the rat-probasin driven MICB transgenic mouse model in our original submission. We have included more description of this mouse model in our revised manuscript. Since there is no human MIC homology in mice, we overexpressed human MICB specifically in mouse prostate directed by the male hormone sensitive promoter, the rat probasin promoter. This is a promoter that commonly used for expressing genes specifically in the prostate . As chronic NKG2D ligand stimulation can down modulate NK and Cd8 T cells function, ectopic expression of MIC under a constitutive promoter, such as EF1a, can resulted in two outcomes: a) suicide impact during embryogenesis due to autoreactive MIC/NKG2D stimulation; b) self-regulatory mechanism to induce NKG2D down-regulation in order to sustain norma embryogenesis as reported ( PMID: 16002667 and reviewed in PMID: 2952534). Therefore, ectopic constitutive overexpression of MIC does not re-capitulate human biology. In order to develop a system that MIC expression is restricted in an organ, we modeled MIC expression specifically in the prostate under the rat probasin promoter. The beauty of this system is that the expression of MIC will be regulated in response to physiological hormone during the puberty of the mice. The immune system of mice will have some experience to the MIC molecule and thus more tolerate than the wild type mice. Indeed, when we inoculate mouse syngeneic tumor cell lines overexpressing MIC into male WT and rPB-MIC mice, the tumor incidence in WT mice was significantly lowered, due to the response to the foreign antigen MIC molecule. We have now included the background in the revised manuscript in the 1st paragraph of the Result Section.

Response to Comment 2:

  1. We agree that tumor microenvironment (TME) plays a significant role in tumor formation/establishment and growth. Thus, to eliminate the variation of tumor microenvironment and to focus on the question that we are investigating, we injected tumors of all three models subcutaneously.
  2. Subcutaneous injection of tumor cell lines, including TRAMP-C2 prostate tumor and LLC lung tumor, is a well-established system for a proof-of-concept study to investigate a specific biology. We do not view the subcutaneous injection skewing our finding; on the contrary, we view it as a system that strengthens our findings by eliminating differential impacts of TME.

Response to Comments 3 and 4:

  1. Thank you for the remarks. We apologize for the typo in Panel C for the number of animals in Figure 2. We have made the corrections in the revised Figures and updated data. We had some unintended visual errors (went crossed-eyed) in initial graphing the individual tumor growth curve in TRAMP-C2 models. For B16-sMICB, we initially performed the experiment with 6 mice which only 3/6 had tumor establishment. In the repeated experiment, we included 12 mice in the experiment, only 6/12 mice had tumor growth. We forgot to update the number of mice. We have made the corrections. All the B16-sMICB mice without tumors showed up as one overlapping single line across the X-axis. In the revised Figure 2, as we are only addressing tumor growth rate of established tumors, we only graphed established tumor growth to avoid confusion. We have clarified this point in the revised Figure legend.

  1. We only calculated the growth rate of established tumors to emphasize our points to differentiate tumor formation vs. tumor growth (or progression).

  1. Tumor growth was calculated by linear regression model using GraphPad Software. We have now included this information in Figure Legend.

Respond to Comment 5.

We have now included the Material and Methods Section of tumor stem cell markers and profiling in the revision.

Respond to Comment 6.

  1. Thank you for the input. In our results, we have provided data of IFNg expression of tumor-infiltrated NK and CD8 T cells in their ability to respond to PMA/I stimulation, which is a simple and ultimate 1st step to test the ability or machinery of NK cell and CD8 T cell response to a generic stimulation. If NK and CD8 T cells have impaired ability to respond to MA/I stimulation, it is unlikely that they will respond well to tumor-specific stimuli. In the revised manuscript, we provided representative and corroborative data of tumors NK and CD8 T cells IFNg and granzyme expression in TRAMP-C2 and TRAMP-C2-sMICB tumors without PMA/I stimulation (Figure S3). Because of these rationales, we did not further analyze tumoral NK and CD8 T cell function in B16 and LLC model without PMA/I stimulation. We did not view these elaborative experiments will provide us more insights into the biology.

  1. Technically, there are very few NK cells in tumors, although very powerful. Thus it is a rather challenging task to isolate tumor infiltrated NK cells for in vitro cytotoxicity assay. Although there are a lot more CD8 T cells in the tumor infiltrated, but those are heterogenic CD8 T cells. In vitro cytotoxicity assay will not be antigen-specific, but limited to CD3/CD28 stimulation, which will not provide more information than what we have presented.

Respond to Comment 7:

We have now included improved Flow cytometry images.

Respond to comment 8:

Thank you for the remark. We apologize for the confusion. We have made a correction to the statement throughout the revised manuscript. To clarify, we sorted for CD44 and CD166 double-positive. The sorted cells were further examined for SCA-1 by flow cytometry analysis. All sorted CD44+CD166+ cells were confirmed to be SCA-1+ as well. We have made corrections and clarifications in the revised manuscript.

Respond to comments 9 and 10:

We have performed manual proof-reading of the revised manuscript. We believe that the writing has been significantly improved in the revision.

Round 2

Reviewer 2 Report

Comments and Suggestions for Authors

I accept the manuscript in the present form.

Reviewer 3 Report

Comments and Suggestions for Authors

The authors have addressed most concerns. For my previous comment 2, the authors misunderstood the point and addressed more on the subcutaneous injection part. 

For most lung cancer model, lung cancer cells are injected into lung/lung parenchyma so that the lung tumor cells are growing in the appropriate microenvironment, and this is more representative for tumor tumor analyses.

In the current study, the authors injected the lung cancer subcutaneously without addressing where the subcutaneous site(s) is (the right flank region seems far away from the lungs) and the concern of inappropriate microenvironment for downstream analyses remains. One would imagine that breast tumor cells would show completely different phenotypic traits when they are injected and grown on the legs (a completely different tissue microenvironment). This is the point I would like to know. However representative of this study would be if the lung/ prostate cancer cells are injected into a completely irrelevant microenvironment and the tumor cells never behave the same as they do naturally. 

"to eliminate the variation of tumor microenvironment and to focus on the question that we are investigating, we injected tumors of all three models subcutaneously."

All types of cancers are different due to their specific microenvironment and thus treatments for each tumor differ. The value of the in vivo model systems is to investigate the tumor cells growth in their natural environment with proper influences of endocrines, metabolites, stromal interactions, cytokines, etc. Therefore, I highly doubt this is anyway reproducible in real human diseases. It is more like an advanced in vitro system.

Comments on the Quality of English Language

N/A

Author Response

Please see the attached response.

Round 3

Reviewer 3 Report

Comments and Suggestions for Authors

The inclusion of new ref supports my previous concern. Although the model design is not ideal, I have no more critique. 

Comments on the Quality of English Language

N/A